# Altitude and SARS-CoV-2 Infection in the First Pandemic Wave in Spain

**DOI:** 10.3390/ijerph18052578

**Published:** 2021-03-04

**Authors:** Jesús Castilla, Ujué Fresán, Camino Trobajo-Sanmartín, Marcela Guevara

**Affiliations:** 1Instituto de Salud Pública de Navarra, 31003 Pamplona, Spain; ujuefresan@gmail.com (U.F.); mguevare@navarra.es (M.G.); 2CIBER Epidemiología y Salud Pública (CIBERESP), 28029 Madrid, Spain; 3Navarre Institute for Health Research (IdiSNA), 31008 Pamplona, Spain; camino.trobajo.sanmartin@navarra.es; 4Instituto de Salud Global (ISGlobal), 08036 Barcelona, Spain; 5Complejo Hospitalario de Navarra, 31008 Pamplona, Spain

**Keywords:** SARS-CoV-2, COVID-19, altitude, seroepidemiological studies, infection transmission, population density

## Abstract

After the first pandemic wave, a nationwide survey assessed the seroprevalence of SARS-CoV-2 antibodies in Spain and found notable differences among provinces whose causes remained unclear. This ecological study aimed to analyze the association between environmental and demographic factors and SARS-CoV-2 infection by province. The seroprevalence of SARS-CoV-2 antibodies by province was obtained from a nationwide representative survey performed in June 2020, after the first pandemic wave in Spain. Linear regression was used in the analysis. The seroprevalence of SARS-CoV-2 antibodies of the 50 provinces ranged from 0.2% to 13.6%. The altitude, which ranged from 5 to 1131 m, explained nearly half of differences in seroprevalence (*R*^2^ = 0.47, *p* < 0.001). The seroprevalence in people residing in provinces above the median altitude (215 m) was three-fold higher (6.5% vs. 2.1%, *p* < 0.001). In the multivariate linear regression, the addition of population density significantly improved the predictive value of the altitude (*R*^2^ = 0.55, *p* < 0.001). Every 100 m of altitude increase and 100 inhabitants/km^2^ of increase in population density, the seroprevalence rose 0.84 and 0.63 percentage points, respectively. Environmental conditions related to higher altitude in winter–spring, such as lower temperatures and absolute humidity, may be relevant to SARS-CoV-2 transmission. Places with such adverse conditions may require additional efforts for pandemic control.

## 1. Introduction

The first wave of the SARS-CoV-2 pandemic spread in Europe in March and April 2020, showing important geographical differences whose causes remain unclear [1]. The role of environmental and demographic conditions on SARS-CoV-2 transmission is not well known. Geographical comparisons have associated higher atmospheric temperature and humidity with lower SARS-CoV-2 transmission [2,3,4,5]. However, these studies may be affected by differences in the completeness of COVID-19 reporting in the first wave, because a high and variable percentage of cases were not confirmed [5]. Furthermore, those studies that compare countries may also be affected by the differences in the introduction of preventive interventions [6,7].

Altitude above sea level could be a potential factor related to SARS-CoV-2 transmission and COVID-19 severity [8]. Studies in Bolivia and Ecuador have suggested that physiological adaptation to hypoxia in high-altitude regions could be protective for SARS-CoV-2 infection [9,10]. However, other authors have also shown that case-fatality rate may not be reduced at high altitude [11]. Indeed, higher mortality rates attributable to COVID-19 have been reported in areas located over 2000 m of altitude against those located below 1500 m in the United States and Mexico [12]. 

Several studies support the hypothesis that low temperature and low absolute humidity favor SARS-CoV-2 transmission. In Italy, a lower risk of hospitalization by COVID-19 was observed in cases in people living near the coast [13].

Spain is a mountainous and climatically diverse territory. Compared to other European countries, the altitude in Spain is higher on average and the geographical diversity is wider, resulting in a major climatic contrast. However, because the altitude range of the Spanish municipalities does not exceed 1700 m, its relevance as a cause of hypoxic effects is low.

Spain was one of the European countries first and most severely affected by COVID-19 [1,14]. Transmission increased steeply during the first half of March, and the main intervention was a lockdown implemented simultaneously in all Spanish regions on 15 March 2020 [15]. No other relevant intervention was introduced; therefore, the important geographical differences observed in COVID-19 cumulative incidence in the first pandemic wave were probably related to case reporting, and to environmental and socio-demographical factors [2,3,4,5,6,7,8,9,10,11,12,13].

This study aimed to assess the possible influence of several environmental and demographic factors on the differences in the transmission of SARS-CoV-2 among provinces in Spain.

## 2. Materials and Methods

This ecological study analyzed environmental and demographic factors to explain the geographical differences among provinces in the seroprevalence of SARS-CoV-2 antibodies after the first pandemic wave in Spain.

Spain is divided into 50 provinces whose capital city is usually the most populated municipality. A nationwide Seroepidemiological Survey of SARS-CoV-2 Infection was performed in a representative sample of the population [16]. The first round of this survey included a representative sample of persons of each province who were recruited from 27 April to 11 May 2020. Immunoassays were performed for the detection of IgG against SARS-CoV-2 in 51,958 persons. The seroprevalence of IgG antibodies was estimated by province [16]. 

Province population data by January 2020 were obtained from the National Institute for Statistics. The Geographical Institute was consulted to obtain the province extension, as well as the latitude and altitude above sea level of their respective capital city (https://www.ign.es/web/ign/portal/ane-datos-geograficos/-/datos-geograficos/datosPoblacion?tipoBusqueda=capitales, accessed date 21 January 2021). Population density was calculated as inhabitants per km^2^.

The increasing part and the peak of the pandemic wave took place in March 2020, and the lockdown started on March 15 and continued up to May; therefore, the meteorological conditions in March were considered relevant to evaluate their potential effect on SARS-CoV-2 transmission. From the National Meteorological Agency, we obtained the monthly mean of each capital city for relative humidity, as well as daily maximum, minimum and mean temperatures in degrees Celsius (°C) (https://opendata.aemet.es/centrodedescargas/productosAEMET?, accessed date 21 January 2021). Absolute humidity was calculated as a function of relative humidity, mean temperature, and atmospheric pressure.

Quantitative variables were presented as the mean, standard deviation, median, and range. Means were compared by the Student’s *t*-test. Categorical variables were presented as percentages and compared by *χ*^2^. Linear regression analysis was used to test the predictive value of each variable on the seroprevalence by province. The determination coefficient (*R*^2^) was calculated as an indicator of the proportion of the seroprevalence variability that was explained by the model. A multivariate linear regression model was fitted, starting with the most predictive variable and adding the variables that significantly increased the predictive capacity of the model. *p*-values lower than 0.05 were considered as statistically significant. The IBM-SPSS Statistics 25 package was used for statistical analysis.

## 3. Results

The seroprevalence of SARS-CoV-2 IgG antibodies of the 50 Spanish provinces ranged from 0.2% to 13.6%. Variability among capital cities was considerable in altitude (ranging from 5 to 1131 m) and population density (ranging from 9 to 840 inhabitants/km^2^). Very important differences were also observed in the average maximum, minimum, and mean temperatures in March 2020, as well as in average atmospheric absolute humidity (Table 1). 

In bivariate analyses, a higher altitude, a lower absolute humidity, and lower averages of daily maximum, minimum and mean temperatures were statistically significantly associated with a higher seroprevalence in the province. Living in a province with a coast was also significantly associated with an average decrease in seroprevalence of 4.1 percentage points (*p* < 0.001; *R*^2^ = 0.30) (Figure 1). However, altitude was the strongest predictive variable for the seroprevalence (*R*^2^ = 0.47) (Table 2 and Figure 2).

In the multivariate linear regression model, population density was the only variable that significantly improved the predictive value of the altitude (*R*^2^ = 0.55). Each 100 m increase in altitude and 100 inhabitants/km^2^ of increase in density, the seroprevalence increased by 0.84 and 0.63 percentage points, respectively (Table 2).

The 25 provinces with altitude above the median (215 m) showed a three-fold higher seroprevalence (6.5% vs. 2.1%, *p* < 0.001), lower absolute humidity (6.4 vs. 8.3 g/m^3^, *p* < 0.001), and lower averages of the daily maximum, minimum and mean temperatures than the others (15.5 vs. 18.6, 4.3 vs. 9.6, 9.9 vs. 13.8 °C, respectively, *p* < 0.001) (Table 3).

## 4. Discussion

Nearly half (47%) of the important geographical differences in SARS-CoV-2 infection during the first pandemic wave in Spain may be explained by the altitude of the province of residence, and this proportion increased up to 55% when population density was also considered. The seroprevalence of SARS-CoV-2 antibodies was three-fold higher in people living in provinces with altitude above the median. The hypothesis derived from these results is that environmental factors may have a role in SARS-CoV-2 transmission greater than previously considered. More studies are needed to evaluate the role of these factors in geographical differences observed in the pandemic. These environmental conditions should be considered in the comparison of the effects of preventive interventions.

It has been suggested that geographical differences in the cumulative incidence of COVID-19 could be indicative of the success of preventive measures implemented [6,7]. However, in the first wave in Spain, a decisive part of SARS-CoV-2 transmission happened before the relevant preventive measures had been introduced, and the lockdown was implemented simultaneously in all regions. Spain has coastline almost all around the country and many mountain ranges. Some provinces in close proximity have notable altitude differences, while some distant provinces have similar altitude; this geographical pattern seems to have carried over to SARS-CoV-2 infection [16].

Bivariate analysis highlighted several environmental variables associated to the higher seroprevalence of SARS-CoV-2 antibodies, including a higher altitude, lower absolute humidity, and lower averages of daily maximum, minimum and mean temperatures, as well as living in a province with a coast. In the present study, the altitude above sea level was more predictive than any meteorological parameter. In Spain in March, those provinces with a higher altitude had, on average, lower absolute humidity, lower temperature in winter, and less sea influence, and these three factors have been related to increases in respiratory virus transmission [2,3,4]. Therefore, the strongest association between higher altitude and SARS-CoV-2 seroprevalence may be explained because this factor combines the effects of absolute humidity, temperature, and proximity to the coast. Environmental factors may modify the efficiency of SARS-CoV-2 transmission; however, they are not sufficient to stop the spread. Therefore, while preventive interventions are necessary for all regions [5,7,17], places with adverse conditions may have additional difficulties for the effective control of transmission and may need more strict preventive measures to achieve similar results.

Other ecological studies have evaluated the relationship between environmental factors and SARS-CoV-2 infection, and many have found an inverse relationship between air temperature and humidity and SARS-CoV-2 infection [2,12,13,18,19,20,21,22]. These studies have been based on COVID-19 reported cases that may be affected by differences in the completeness of reporting. In the first pandemic wave, a high percentage of cases were not confirmed, and this percentage may be different among regions and countries [5]. Furthermore, those studies that compare countries may be also affected by the differences in the introduction of preventive interventions [6,7]. Our results are consistent with most of these studies, and additionally have the advantage of the optimal comparability among Spanish provinces because seroprevalence was assessed with the same protocol, interventions were simultaneously introduced, and social differences within the same country are probably smaller.

Studies in Latin America have not found altitude to be associated with higher transmission [16,17,18], but the wider altitude range and the tropical latitude make comparisons with our results difficult.

We found a lower seroprevalence of SARS-CoV-2 antibodies in provinces with a coast. This result might be consistent with the lower risk of hospitalization among COVID-19 cases in people living near the coast, which has been found in Italy [13].

The results of our bivariate analysis are consistent with worldwide and large countries’ studies that did not find association between population density and SARS-CoV-2 spread [2,20]. However, a higher population density in the province was an independent predictor of SARS-CoV-2 seroprevalence when altitude was an adjusted factor. Although population density was not an important factor in COVID-19 spreading under strict lockdown policies [23], it has been also described as a relevant factor for SARS-CoV-2 transmission in less strict conditions [24,25,26,27]; therefore, some effect of population density on transmission in the period before the lockdown in Spain seems plausible.

The main strengths of this study are that Spain met special conditions in the first pandemic wave to evaluate this association, because it is an environmentally diverse territory, the lockdown was implemented simultaneously in all regions, and a nationwide survey was performed to obtain comparable estimates of the seroprevalence of SARS-CoV-2 by provinces.

This study has limitations, however. The ecological analysis may suggest a hypothesis but cannot conclude causality. Parameters of the capital city were assigned to the whole province; although, in most provinces, parameters of the capital city are a good proxy of the average conditions to which the population of the province is exposed. These results may not predict infections in further waves because a part of the population may have acquired immunity; many preventive interventions have been introduced; and the weather conditions in other months may be different. Social interaction and differences between rural and urban residence have been suggested as important factors for SARS-CoV-2 transmission; however, population density alone was not predictive for higher seroprevalence in the present study. Several potential risk factors have not been considered in the analysis, but the strong association that we found between altitude and seroprevalence is unlikely to be totally explained by confounding factors. Furthermore, because altitude is a primary factor, other environmental and social variables might act as intermediate factors explaining the association between altitude and SARS-CoV-2 infection.

## 5. Conclusions

In summary, important differences among Spanish provinces were observed in the seroprevalence of SARS-CoV-2 antibodies after the first pandemic wave. Nearly half of these differences may be explained by altitude. Environmental conditions related to higher altitude in winter–spring, such as the combination of low temperature and absolute humidity, may be relevant for SARS-CoV-2 transmission. A higher population density was also a predictive factor for transmission. Although spread of SARS-CoV-2 widely depends on social behaviors and preventive interventions, places with adverse environmental conditions may require additional efforts to achieve similar pandemic control.

## Figures and Tables

**Figure 1 ijerph-18-02578-f001:**
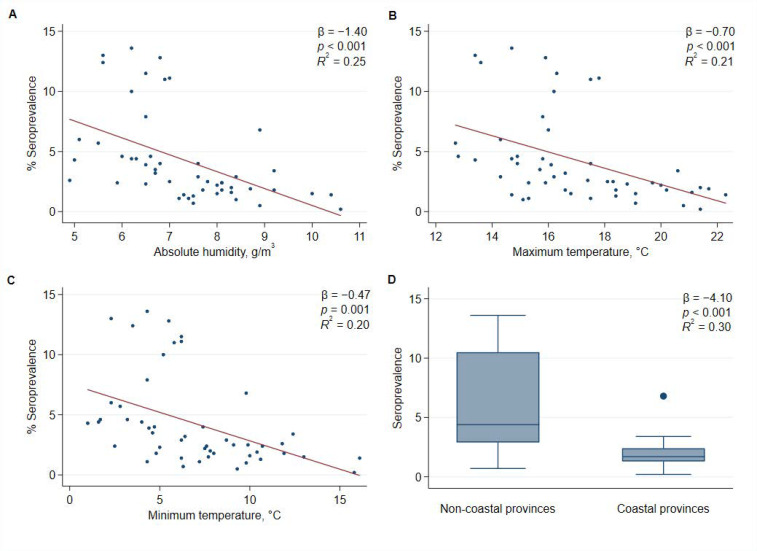
Bivariate association of the seroprevalence of IgG antibodies against SARS-CoV-2 as a function of four environmental parameters by province. The seroprevalence was obtained by immunoassay tests in samples collected from 27 April to 11 May 2020, in Spain. (**A**) Average of absolute humidity; (**B**) average of daily maximum temperatures; (**C**) average of daily minimum temperatures; and (**D**) box-plot of comparison between coastal and non-coastal provinces.

**Figure 2 ijerph-18-02578-f002:**
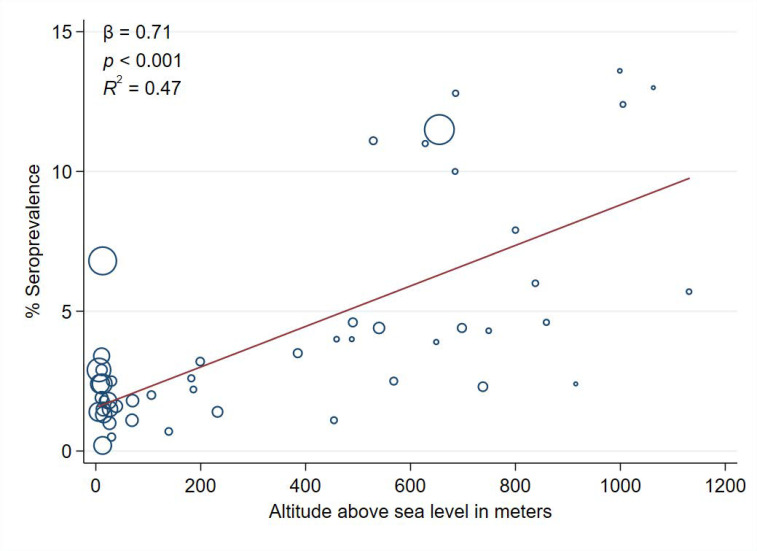
Linear regression analysis of the seroprevalence of IgG antibodies against SARS-CoV-2 as a function of the altitude above sea level in meters by province. The diameter of the circles is proportional to the population density. The seroprevalence was obtained by immunoassay tests in samples collected from 27 April to 11 May 2020, in Spain.

**Table 1 ijerph-18-02578-t001:** Descriptive analysis of the variables considered for the 50 provinces in, Spain.

Variables Analyzed	Mean	SD	Range	Median
Seroprevalence of SARS-CoV-2 IgG, % *	4.3	3.8	0.2–13.6	2.8
Altitude above sea level, m **	370	362	5–1131	215
Latitude in degrees **	39.9	3.2	28.1–43.3	40.5
Average absolute humidity (g/m^3^) **	7.3	1.3	4.9–10.6	7.3
Average of daily maximum temperatures **	17.0	2.5	12.7–22.3	16.5
Average of daily minimum temperatures **	7.0	3.6	1.0–16.1	6.6
Average of daily mean temperatures **	11.9	3.0	6.2–19.2	11.7
Population (in thousands)	943	1208	90–6747	643
Extension (km^2^)	9921	4805	1980–21,766	9992
Density of population (inhabitants/km^2^)	132	174	9–840	63

* Seroprevalence of SARS-CoV-2 IgG antibodies obtained by immunoassay tests from 27 April to 11 May 2020. ** Data refer to the capital city of the province. Meteorological variables were average variables of March 2020. Temperature is in degrees Celsius.

**Table 2 ijerph-18-02578-t002:** Results of linear regression at the province level (*n* = 50) between environmental and demographic variables and the seroprevalence of IgG antibodies against SARS-CoV-2 obtained by immunoassay tests from 27 April to 11 May 2020, in Spain.

Analyses and Variables	Beta	SE	*p*-Value	*R* ^2^
**Bivariate analyses**				
Altitude above sea level (per 100 m) *	0.71	0.11	<0.001	0.47
Latitude in degrees *	0.19	0.17	0.261	0.03
Average absolute humidity (g/m^3^) *	−1.40	0.35	<0.001	0.25
Average of daily maximum temperatures, °C *	−0.70	0.19	<0.001	0.21
Average of daily minimum temperatures, °C *	−0.47	0.13	0.001	0.20
Average of daily mean temperature, °C *	−0.55	0.16	0.001	0.19
Province with coast	−4.10	0.90	<0.001	0.30
Density of population (inhabitants per km^2^)	−0.07	0.31	0.821	0.001
**Multivariable analyses**				0.55
Intercept	0.36	0.71	0.615	
Altitude above sea level (per 100 m) *	0.84	0.11	<0.001	
Density of population (inhabitants per km^2^)	0.63	0.23	0.009	

* Data refer to the capital city of the province; *R*^2^, determination coefficient. SE, standard error; We used bold case to distingish the analysis, because normal case is used to variables.

**Table 3 ijerph-18-02578-t003:** Comparison of average characteristics between provinces (*n* = 50) with altitude above and below the median (215 m). Spain, 2020.

Variables Analyzed	Provinces with Altitude >215 m (*n* = 25)		Provinces with Altitude <215 m (*n* = 25)		*p*-Value
	Mean	SD	Mean	SD	
Seroprevalence of SARS-CoV-2 IgG, % *	6.5	4.1	2.1	1.3	<0.001
Latitude in degrees **	40.8	1.7	39.0	4.0	0.043
Average absolute humidity (g/m^3^) **	6.4	0.7	8.3	1.2	<0.001
Average of daily maximum temperatures, °C **	15.5	1.7	18.6	2.3	<0.001
Average of daily minimum temperatures, °C **	4.3	1.9	9.6	2.8	<0.001
Average of daily mean temperatures, °C **	9.9	1.7	13.8	2.7	<0.001
Density of population (inhabitants per km^2^)	69	163	194	164	<0.001

* Seroprevalence of SARS-CoV-2 IgG antibodies obtained by immunoassay tests from 27 April to 11 May 2020. ** Data refer to the capital city of the province. Meteorological variables were average variables of March 2020.

## Data Availability

All data are available from the Spanish Meteorological Agency (https://opendata.aemet.es/centrodedescargas/productosAEMET?, accessed date 21 January 2021), Spanish Geographical Institute (https://www.ign.es/web/ign/portal/ane-datos-geograficos/-/datos-geograficos/datosPoblacion?tipoBusqueda=capitals, accessed date 21 January 2021), and Instituto de Salud Carlos III (https://portalcne.isciii.es/enecovid19/, accessed date 21 January 2021).

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
