# Peer review of "Altitude and SARS-CoV-2 Infection in the First Pandemic Wave in Spain"

_ijerph, 2021, doi:10.3390/ijerph18052578_

Round 1

Reviewer 1 Report

Dear Authors, 

The novelty of your manuscript is quite interesting. I can appreciate that you follow the Journal recommendations and also, overall the article is well-written.

The only question that I have is about other countries with similar geographical climate and environmental factors. You can comment extensively about this comparison. In line with this, you can upgrade your reference list!

line 47. page 2 : The first question that I have is about other countries with similar geographical climate and environmental factors. You can comment extensively about this comparison. In line with this, you can upgrade your reference list!

line 157 , page 5.The the second recommendation is to be more extensive with de discussion part especially regarding the climatic type (temperature, humidity) of these countries.

Author Response

The novelty of your manuscript is quite interesting. I can appreciate that you follow the Journal recommendations and also, overall the article is well-written.

AUTORS RESPONSE: Thank you very much for your time reviewing our manuscript.

The only question that I have is about other countries with similar geographical climate and environmental factors. You can comment extensively about this comparison. In line with this, you can upgrade your reference list!

AUTORS RESPONSE: Nine new references have been added (references 8,9,11,12,13,21,22,23, and 27) .

line 47. page 2 : The first question that I have is about other countries with similar geographical climate and environmental factors. You can comment extensively about this comparison. In line with this, you can upgrade your reference list!

AUTORS RESPONSE: We have added 9 additional references with results from other countries: China, Italy, US, Mexico, Peru, Ecuador. (references 8,9,11,12,13,21,22,23, and 27) .

line 157 , page 5.The the second recommendation is to be more extensive with de discussion part especially regarding the climatic type (temperature, humidity) of these countries.

AUTORS RESPONSE: We have enlarged the discussion and we have compared our results with those from other countries. Lines 185-210, 217-219, and 220-227.

Reviewer 2 Report

The article is very timely and interesting. However, I have the following concerns:

1) Authors do not quote relevant literature, including for example the narrative review by Millet et al. Authors should provide a better start-of-art of the findings, including those contrasting and inconclusive.

2) Also discussion should be improved, better comparing and stressing authors' findings against the literature. 

3) Why only the graph of altitude is provided? Authors should provide also other relevant graphs. 

4) Please provide more details concerning methods.

Author Response

The article is very timely and interesting. However, I have the following concerns:

1) Authors do not quote relevant literature, including for example the narrative review by Millet et al. Authors should provide a better start-of-art of the findings, including those contrasting and inconclusive.

AUTORS RESPONSE: Thank you very much for your time reviewing our manuscript.

The introduction had incorporated the reference of Millet et al as well as other new references. Now, we give a more complete description of the state-of-art. Lines 42-58,

2) Also discussion should be improved, better comparing and stressing authors' findings against the literature. 

AUTORS RESPONSE: The discussion has been enlarged with new comparisons of our results with the literature. Lines 185-210, 217-219, and 220-227.

3) Why only the graph of altitude is provided? Authors should provide also other relevant graphs. 

AUTORS RESPONSE: A new figure 1 with four panels has been added.

4) Please provide more details concerning methods.

AUTORS RESPONSE: More details about the statistical analysis have been added in the methods section. Now we say “A multivariate linear regression model was fitted starting with the most predictive variable and adding the variables that increased significantly the predictive capacity of the model. P values lower than 0.05 were considered as statistically significant. The IBM-SPSS Statistics 25 package was used for statistical analysis.”

Reviewer 3 Report

Major comment:

This study aims to explore the relationship between climate and geological parameters and the seroprevalence of SARS-CoV-2. In general, an ecological study could be helpful in the generation of hypotheses. Therefore, the authors should propose their hypothesis derived from this study and their comments for further research.

Minor suggestion:

  1. Improve the resolution of Figure 1.
  2. The discussion section is too short. Need more in-depth discussion.

Author Response

Major comment:

This study aims to explore the relationship between climate and geological parameters and the seroprevalence of SARS-CoV-2. In general, an ecological study could be helpful in the generation of hypotheses. Therefore, the authors should propose their hypothesis derived from this study and their comments for further research.

AUTORS RESPONSE: Thank you very much for your time reviewing our manuscript.

In the discussion section, we have added a new paragraph with the hypothesis derived from our results and suggesting further implications.

Discussion, first paragraph: “The hypothesis derived from these results is that environmental factors may have a role in SARS-CoV-2 transmission higher than the previously considered. More studies are needed to evaluate the role of these factors in geographical differences observed in the pandemic. These environmental conditions should be considered in the comparison of the effects of preventive interventions.”

Minor suggestion:

  1. Improve the resolution of Figure 1.

AUTORS RESPONSE: The resolution of the figure has been improved.

  1. The discussion section is too short. Need more in-depth discussion.

AUTORS RESPONSE: Now we provide a more in-depth discussion, with new comparisons of our results with the literature. Lines 185-210, 217-219, and 220-227.

Round 2

Reviewer 2 Report

Authors have addressed all comments.